# Predictors of neonatal mortality in Ethiopia: Cross sectional study using 2005, 2010 and 2016 Ethiopian demographic health survey datasets

**Yirgalem Shibiru Baruda**[1]*, **Mark Spigt**[2], **Andrea Gabrio**[3], **Lelisa Fikadu Assebe**[4]

**1** Faculty of Health, Medicine and Life Sciences, Maastricht University, Maastricht, the Netherlands, **2** Department of Family Medicine, Faculty of Health, Medicine and Life Sciences, Maastricht University, Maastricht, the Netherlands, **3** Department of Methodology and Statistics, Faculty of Health, Medicine and Life Sciences, Maastricht University, Maastricht, the Netherlands, **4** Department of Global Public Health and Primary Care, Faculty of Medicine, University of Bergen, Bergen, Norway

* yirgalemsh75@gmail.com

**Data Availability Statement:** The datasets generated and/or analyzed during the current study are publicly available and can access through

## Abstract

Ethiopia is among the countries that have highest neonatal mortality in the world. Despite efforts to reduce neonatal mortality, the country has faced challenges in achieving national and global targets. The study aims to determine the trends and predictors of change in neonatal mortality in Ethiopia for the past 15 years. The study used Ethiopian Demographic Health Survey Datasets (EDHS) from 2005, 2011, and 2016. All live births of reproductive-age women in Ethiopia were included in the study. Multivariate decomposition analysis for the nonlinear response variable (MVDCMP) based on the logit link function was employed to determine the relative contribution of each independent variable to the change in neonatal mortality over the last 15 years. The neonatal mortality rate has decreased by 11 per 1,000 live births, with an annual reduction rate of 2.8% during the survey period. The mortality rate increased in the pastoralist regions of the country from 31 per 1,000 live births to 36 per 1,000 live births, compared to the city and agrarian regions. Maternal ANC visits in 2005 and 2016 (AOR [95%CI] = 0.10 [0.01, 0.81]; 0.01 [0.02, 0.60]) were significantly associated with decreased neonatal mortality. In addition, the decomposition analysis revealed that increased birth interval of more than 24 months and early breastfeeding initiation contributed to the reduction of neonatal mortality by 26% and 10%, respectively, during the survey period. The study found that neonatal mortality is a public health problem in the country, particularly in pastoralist communities. Tailor made maternal and child healthcare interventions that promote early breastfeeding initiation, increased birth intervals and ANC utilization should be implemented to reduce neonatal mortality, particularly in pastoralist communities.

## Introduction

Neonatal mortality is defined as the death of a newborn occurring within the first 28 days of life. It is further classified as early neonatal mortality when the death occurs before the age of

(https://dhsprogram.com/data/new-useregistration.cfm).

**Funding:** The author did not receive any grants for conducting the research, except for the publication fee provided by the University of Bergen.

**Competing interests:** The authors have declared that no competing interests exist.

seven days, and late neonatal mortality, when the death occurs between the ages of seven and twenty-eight days [1]. Moreover, neonatal mortality accounts for nearly half of all under-five mortality, indicating that the neonatal period is the most high-risk time for child survival [2]. The global average annual rate of reduction in neonatal mortality rate was 2.6 percent, which was lower than the 3.6 percent decline in children aged 1–59 months [3]. According to UNICEF, 61 countries will fail to meet the SDG3 target of reducing the mortality rate to 12 or fewer deaths per 1,000 live births by 2030, despite the fact that around 75% of neonatal mortality in LMICs, including sub–Saharan African countries, can be prevented [4].

In addition, almost a third of all neonatal deaths occur during the first day of life and nearly three-quarters occur within the first week [2]. Sub-Saharan Africa has the highest neonatal mortality rate, with 27 deaths per 1,000 live births, in the world [5]. Ethiopia has the highest neonatal mortality, although the neonatal mortality reduced from 49 to 28 per 1,000 live births between 2000 and 2016 [6]. The mortality rate varies across the regions in the country, the Benishangul Gumuz region had the highest mortality rate at 55 per 1,000 live births compared to Addis Ababa city, which had the lowest mortality rate [6].

Poor maternal utilization of health care services during pregnancy and childbirth is considered to be an important factor to the high prevalence of preventable neonatal mortality in LMICs [7]. Previous studies have shown that antenatal care (ANC) visits during pregnancy [8], institutional delivery [9, 10], and postnatal care [11] reduce the risk of neonatal mortality in LMICs. In addition, failure to initiate breastfeeding early, within 1 hour of birth, predisposes the newborns to infection, resulting in an increased risk of neonatal mortality [12, 13]. Similarly, lack of sufficient birth intervals also contributed to inadequate time for mothers to recover from nutritional depletion, leading to poor newborn health outcomes and deaths [14, 15]. The WHO recommends spacing childbirths at least 24–36 months apart [16]. Moreover, previous studies conducted in Angola [17], Uganda [18] and other LMICs [19] have shown that place of residence, maternal age at first birth, birth weight, household wealth index, and maternal education were all significantly associated with neonatal mortality.

Previous studies have shown the influence of several factors in alleviating neonatal mortality in Ethiopia [20–22]. However, little evidence is available about the factors' contribution to the reduction of neonatal mortality over the past decades. Thus, this study aims to examine the contribution of these factors to the change in neonatal mortality over time in Ethiopia.

## Methods

### Study area and setting

Ethiopia is located in the Horn of Africa and, after Nigeria, it is Africa's second-most populous country. It had 115 million people in 2020 and is expected to grow at a 3.09 percent annual rate [23]. Children under 15 make up 47% of the population, while people aged 15 to 65 make up 49%. The sex ratio between males and females is almost equal, and women of reproductive age constitute about 23% of the population [24]. Nearly 80% of the population live in rural areas and depend mainly on subsistence agriculture. Also, the agriculture sector employed around 70% of the population [25].

### Study design and data source

Cross-sectional studies were conducted in Ethiopia in 2005, 2010, and 2016. The surveys collected nationally representative data. The data was collected by the Central Statistical Agency (CSA) in collaboration with the Ethiopian Public Health Institute (EPHI) and the Minister of Health (MoH). This study used the subsamples of the 2005, 2011, and 2016 EDHS datasets.

**Study population and sample.** The sampling in the EDHS was a stratified, two-stage sampling procedure to recruit a nationally representative sample for the surveys. Each region was stratified into urban and rural areas, resulting in enumeration areas (EAs), the sampling stratum for the surveys. In the first stage, 645 EAs in EDHS 2005, 624 EAs in EDHS 2011, and 645 EAs in EDHS 2016 were randomly selected proportional to their EA size, and on average, 27 to 32 households per EA were chosen in the second stage. The sample unit for the reported study was all live births reported from each survey. As a result, the birth recodes (BR) 2005, 2010, and 2016 datasets were used for the analysis because they contain birth history data of interviewed women, including pregnancy, delivery, and post-natal care, as well as nutrition and health data for newborns in the last five years. A weighted sample of 33,976 (11,163 in the EDHS 2005, 11,872 in the EDHS 2011, and 10,941 in the EDHS 2016) live births who were born in the five years preceding the surveys were considered for the analysis.

## Study variables and measures

The dependent variable for the study, neonatal mortality, was classified dichotomously as "1 or 0" 1 death of a child at the ages 0 to 28 days and 0 if the child was alive. The following independent variables were coded as categorical nominal variables: ANC visit (yes or no), mode of delivery (vaginal or cesarean delivery), place of delivery (home or health institution, other) and early initiation of breastfeeding (yes and no), maternal marital status (never married, married, divorced/widow); maternal religion (orthodox, muslim, protestant, catholic, and others), place of residence (urban and rural), the regions of residence were categorized based on the living habits according to the ministry of health's classification: agrarian (Tigray, Amhara, Oromia, SNNPR), pastoralist (Somali, Afar, Benishangul, Gambella), and city (Harari, Addis Ababa, and Dire Dawa) [26] In addition, other independent variables were recoded as ordinal categorical variables: birth size (large, appropriate, small); maternal age at first birth (<20, 20–29, 30–39, 40–49 years), birth order (coded as 1, 2–4, $\geq$ 5), birth interval (<24 or $\geq$24 months), maternal education (no education, primary, and secondary and higher) and household wealth index (coded as poor, middle and rich).

## Statistical analysis

Data weighting was done to adjust the representativeness of the different regions in the survey and to obtain reliable statistical estimates. Descriptive statistics of independent and dependent variables were presented as frequency distributions and percentages. In addition, a multi-collinearity check was carried out using collinearity diagnostics with the variance inflation factor. The datasets were appended using the extracted data from 2005, 2011, and 2016, after maintaining similar variables across the surveys, to do trend analysis for neonatal mortality. The statistical analysis was done using the svyset (analysis plan) command to adjust the analysis for multi-stage sampling. Sampling weight variable, primary sampling unit, and strata were used to create the svyset command. STATA software version 17 was used to analyze the data.

A multivariate decomposition analysis for the nonlinear response variable (MVDCMP) based on the logit link function was employed to identify the relative contribution of each independent variable to the change in neonatal mortality over the last 15 years. The analysis model is defined as a logistic regression model for neonatal mortality where the estimated logit of mortality for neonate i in year j:

$$Y = F(X\beta) = \text{logit}(Y\_ij) = X\_ij \; \beta\_j$$

where logit refers to logistic function; β is the regression coefficient on the logit scale of each selected independent variable X [27]. Using this model, the proportion difference in Y between

$$Logit(Y_{2005}) - Logit(Y_{2016}) = F(X_{2005}\beta_{2016}) - F(X_{2016}\beta_{2016})$$

$$= \underbrace{F(x_{2005}\beta_{2005}) - F(X_{2016}\beta_{2005})}_{E} + \underbrace{F(X_{2016}\beta_{2005}) - F(X_{2016}\beta_{2016})}_{C}$$

**Fig 1. Compositional differences between groups and differences in the effects of characteristics between groups.** E represents a counterfactual comparison of the change in neonatal mortality from group 2005's perspective (that is, the anticipated change in neonatal mortality if survey 2005 was given survey 2016's distribution of Xs). C is a counterfactual comparison of neonatal mortality from survey 2016's perspective (the predicted change in neonatal mortality if survey 2016 experienced survey 2005's characteristics pattern or effect (β) to X).

survey in year j = 2005 and survey in year j = 2016 of successive EDHS surveys for the neonatal mortality can be shown to be decomposed as (Fig 1).

## Data access and ethical consideration

The dataset was officially accessed on March 31, 2022, after a request outlining the purpose of the study had been submitted to the Demographic Health Survey Program (https://dhsprogram.com/data/new-use-registration.cfm). In addition, ethical clearance was secured from the ethical committee from the Faculty of Health, Medicine, and Life Sciences at Maastricht University.

## Results

### Socio-demographic characteristics of children

The socio-demographic characteristics of births in the five years preceding the survey are shown in Table 1. A total of 33,976 live births were included in the study. The geographical distribution of the births across the surveys remained similar; more than 92% of the live births were from rural and agrarian regions. Also, the Oromia region, specifically, accounted for 40% of live births. Regarding the mothers' educational status, the percentage of mothers who could not read and write decreased from 79.1% in 2005 to 66% in 2016. Similarly, mothers' secondary and higher education levels increased steadily from 4.2% in 2005 to 7.2% in 2016. In all surveys, almost 50% of the mothers were aged between 20 and 29 years.

### Trend and difference in neonatal mortality by the predictors

In this section, the trends in neonatal mortality are examined over time for each predictor and the decline in neonatal mortality for each predictor is assessed between 2011 and 2005, 2016 and 2011, 2016 and 2005.

As shown in Table 2, the mortality rate increased among children born in the pastoralist region between 2005 and 2016, from 31 per 1,000 live births to 36 per 1000 live births, but declined in city and agrarian regions. Similarly, neonatal mortality fell from 39 per 1,000 live births to 30 per 1,000 live births among children born to mothers with no education and from 44 per 1,000 live births to 24 per 1,000 live births among those with primary education between 2005 and 2016.

Table 3 displays that in all surveys, neonatal mortality was higher among children born to mothers who delivered their first child under 18 than mothers older than 18. Similarly, the mortality was significantly higher (more than five times) among children with multiple births, two or more birth, compared to single births. The mortality rate was twice as high for

**Table 1. Frequency and percentage of the respondents and their children socio–demographic characteristics among children born in the five years preceding the survey, EDHS 2005, 2011 and 2016.**

| Variables | | EDHS2005 (N = 11,163) | EDHS 2011 (N = 11,872) | EDHS 2016 (N = 10,941) |
|---|---|---|---|---|
| | | Weighted N (%) | Weighted N (%) | Weighted N (%) |
| Regions | Tigray | 698(6.3) | 753 (6.3) | 710 (6.5) |
| | Afar | 107(1.0) | 121(1.0) | 114 (1.0) |
| | Amhara | 2,621(23.5) | 2,656 (22.4) | 2,054 (18.8) |
| | Oromia | 4,411(39.5) | 5,014 (42.2) | 4,817 (44.0) |
| | Somali | 477(4.3) | 364 (3.1) | 507 (4.6) |
| | Benishangul | 105(0.9) | 140 (1.2) | 121 (1.1) |
| | SNNPR | 2,500 (22.4) | 2,494 (21.0) | 2,278 (20.8) |
| | Gambela | 31(0.3) | 40 (0.3) | 27 (0.2) |
| | Harari | 22(0.2) | 29 (0.2) | 26 (0.2) |
| | Addis Ababa | 153(1.4) | 222 (1.9) | 241 (2.2) |
| | Dire Dawa | 37(0.3) | 39 (0.3) | 47 (0.4) |
| Regional | Agrarian | 10,229 (91.6) | 10,916 (92.0) | 9,859 (90.1) |
| Context | pastoralist | 721 (6.5) | 666 (5.6) | 768 (7.0) |
| | City | 213 (1.9) | 289 (2.4) | 314(2.9) |
| Place of residence | Urban | 815(7.3) | 1,528 (12.9) | 1,205 (11.0) |
| | Rural | 10,348 (92.7) | 10,344 (87.1) | 9,736 (89.0) |
| Religions | Orthodox | 4,674 (41.9) | 4,519 (38.1) | 3,742 (34.2 |
| | Catholic | 121 (1.1) | 108 (0.9) | 103 (0.9) |
| | Protestant | 2,217 (19.9) | 2,758 (23.2) | 2,314 (21.2) |
| | Muslim | 3,875 (34.7) | 4,214 (35.5) | 4,532 (41.4) |
| | Traditional | 173 (1.6) | 124 (1.0) | 142 (1.3) |
| | Other | 102 (0.9) | 142 (1.2) | 107 (1.0) |
| Educational attainment | No education | 8,838 (79.2) | 8227 (69.3) | 7,221 (66.0) |
| | Primary education | 1,855 (16.6) | 3,211 (27.1) | 2,937 (26.9) |
| | Secondary and higher education | 470 (4.2) | 434 (3.7) | 782 (7.2) |
| Wealth Status | Poor | 4,796 (43.0) | 5,368 (45.2) | 5,111 (46.7) |
| | Middle | 2,486 (22.3) | 2437 (20.5) | 2263 (20.7) |
| | Rich | 3,882 (34.8) | 4067 (34.3) | 3567 (32.6) |
| Mother's age | <20 | 575 (5.2) | 492 (4.1) | 378 (3.5) |
| | 20–29 | 5,415 (48.5) | 6,158 (51.9) | 5,388 (49.3) |
| | 30–39 | 4,019 (36.0) | 4,166 (35.1) | 4,221 (38.6) |
| | ≥40 | 1,154 (10.3) | 1,057 (8.9 | 953 (8.7) |
| Current marital Status | Currently Married or Union | 1,0518 (94.2) | 10,989 (92.6) | 10,381 (94.9) |
| | not currently Married or Union | 645 (5.8) | 883 (7.4) | 560 (5.1) |

newborns from mothers who had a birth interval of less than 24-months. Neonatal mortality was also significantly lower among newborns who were breastfed within the first hour of life; 2005, 13 per 1000 live birth, and in 2016, 6.5 per 1000 live birth.

There was no difference in neonatal mortality among children whose mothers had no ANC visits between 2005 and 2016. However, the mortality rate was reduced among children born to mothers who received ANC visits in the same period. Also, the mortality rate increased among children delivered through cesarean delivery compared to those vaginally delivered between 2005 and 2016 (Table 4).

**Table 2. Neonatal mortality rate (NMR) (five-year rate) and the difference in NMR between surveys by socio- demographic characteristics of the respondents, EDHS 2005, 2011 and 2016.**

| Variables | | EDHS 2005 | EDHS 2011 | EDHS 2016 | 2011–2005 | 2016–2011 | 2016–2005 |
|---|---|---|---|---|---|---|---|
| | | NMR | NMR | NMR | NMR difference between surveys | | |
| Regions | Tigray | 26.9 | 41.3 | 27.1 | 14.4 | -14.2 | 0.2 |
| | Afar | 29.2 | 18.9 | 26.4 | -10.3 | 7.5 | -2.8 |
| | Amhara | 52.1 | 46.5 | 32.4 | -5.6 | -14.1 | -19.7 |
| | Oromia | 38.5 | 34.1 | 27.7 | -4.4 | -6.4 | -10.8 |
| | Somali | 30.1 | 28.7 | 40.9 | -1.4 | 12.2 | 10.8 |
| | Benishangul | 37.9 | 47.9 | 28.8 | 10 | -19.1 | -9.1 |
| | SNNPR | 33.6 | 35.4 | 24.8 | 1.8 | -10.6 | -8.8 |
| | Gambela | 23.6 | 36.5 | 28.2 | 12.9 | -8.3 | 4.6 |
| | Harari | 22.2 | 40.8 | 31.5 | 18.6 | -9.3 | 9.3 |
| | Addis Ababa | 28.5 | 21.7 | 21.0 | -6.8 | -0.7 | -7.5 |
| | Dire Dawa | 26.9 | 16.4 | 30.0 | -10.5 | 13.6 | 3.1 |
| Regional Context | Agrarian | 39.9 | 37.9 | 27.9 | -2 | -10 | -12 |
| | pastoralist | 30.8 | 31.3 | 36.3 | 0.5 | 5 | 5.5 |
| | City | 27.5 | 22.8 | 23.1 | -4.7 | 0.3 | -4.4 |
| Place of residence | Urban | 43.7 | 41.3 | 34.1 | -2.4 | -7.2 | -9.6 |
| | Rural | 38.8 | 36.6 | 27.7 | -2.2 | -8.9 | -11.1 |
| Religion | Orthodox | 42.2 | 42.9 | 31.0 | 0.7 | -11.9 | -11.2 |
| | catholic | 33.8 | 81.8 | 5.8 | 48 | -76 | -28 |
| | protestant | 37.1 | 28.9 | 24.0 | -8.2 | -4.9 | -13.1 |
| | Muslim | 37.9 | 35.5 | 30.1 | -2.4 | -5.4 | -7.8 |
| | traditional | 16.5 | 4.8 | 3.0 | -11.7 | -1.8 | -13.5 |
| | other | 33.7 | 61.4 | 16.7 | 27.7 | -44.7 | -17 |
| Educational attainment | No Edu | 38.8 | 38.1 | 29.8 | -0.7 | -8.3 | -9 |
| | Primary | 44.7 | 35.7 | 24.1 | -9 | -11.6 | -20.6 |
| | Sec and higher | 23.8 | 31.4 | 31.6 | 7.6 | 0.2 | 7.8 |
| Wealth status | poor | 30.7 | 44.7 | 22.5 | 14 | -22.2 | -8.2 |
| | middle | 55.3 | 29.9 | 31.0 | -25.4 | 1.1 | -24.3 |
| | rich | 39.2 | 31.7 | 35.1 | -7.5 | 3.4 | -4.1 |
| Mother's age | <20 | 58.0 | 78.8 | 18.0 | 20.8 | -60.8 | -40 |
| | 20–29 | 44.4 | 36.1 | 26.5 | -8.3 | -9.6 | -17.9 |
| | 30–39 | 27.0 | 33.3 | 29.1 | 6.3 | -4.2 | 2.1 |
| | ≥40 | 47.5 | 39.8 | 40.3 | -7.7 | 0.5 | -7.2 |
| Current Marital Status | Currently Married or Union | 38.1 | 36.8 | 28.4 | -1.3 | -8.4 | -9.7 |
| | not currently Married or Union | 57.0 | 42.1 | 28.9 | -14.9 | -13.2 | -28.1 |

## Binary logistic regression on socio-demographic, health seeking, fertility and nutritional predictors on neonatal mortality

The multivariable logistic regression analysis shows those who were born in the cities, were less likely to die in their first month of life than children born in agrarian and pastoralist regions in 2011 (AOR [95%CI] = 0.11[0.02, 0.46]). Newborns whose mothers had five or more ANC visits were also less likely to die in the first month in 2005 and 2016 (AOR [95%CI] = 0.10[0.01, 0.81]; AOR [95%CI] = 0.01[0.02, 0.60]) than children from mothers who did not have ANC visits. Likewise, compared to children whose mothers had no visits, those whose mothers had one to four ANC visits were also less likely to die in the first 28 days in 2016 (AOR [95%CI] = 0.38[0.16, 0.89]). Neonatal mortality was lower among children born with a

**Table 3.** Neonatal mortality rate (NMR) (five-year rate) and the difference in NMR between surveys by selected child characteristics and nutritional factors in 2005, 2011 and 2016 EDHS.

| | | EDHS 2005 | EDHS 2011 | EDHS 2016 | 2011–2005 | 2016–2011 | 2005–2016 |
|---|---|---|---|---|---|---|---|
| **Variables** | | NMR | NMR | NMR | NMR difference between surveys | | |
| Age at first birth | <18 | 46.4 | 38.8 | 34.8 | -7.6 | -4 | -11.6 |
| | 18–34 | 33.5 | 35.7 | 24.2 | 2.2 | -11.5 | -9.3 |
| | ≥35 | 0.0 | 209.4 | 0.0 | 209.4 | -209.4 | 0 |
| Type of birth | Single birth | 36.6 | 34.0 | 25.3 | -2.6 | -8.7 | -11.3 |
| | Twin birth | 181.2 | 172.9 | 142.7 | -8.3 | -30.2 | -38.5 |
| Preceding birth interval | <24mths | 55.7 | 39.1 | 50.2 | -16.6 | 11.1 | -5.5 |
| | ≥24mths | 26.6 | 32.4 | 24.4 | 5.8 | -8 | -2.2 |
| Sex of child | Male | 45.6 | 45.3 | 39.4 | -0.3 | -5.9 | -6.2 |
| | female | 32.4 | 28.4 | 16.5 | -4 | -11.9 | -15.9 |
| Birth order | 1 | 59.8 | 46.0 | 22.9 | -13.8 | -23.1 | -36.9 |
| | 2–4 | 31.2 | 37.2 | 26.7 | 6 | -10.5 | -4.5 |
| | ≥5 | 38.7 | 32.7 | 33.0 | -6 | 0.3 | -5.7 |
| Early initiation of BF | yes | 13.0 | 11.9 | 6.5 | -1.1 | -5.4 | -6.5 |
| | no | 15.3 | 6.7 | 9.3 | -10.6 | 3.4 | -7.2 |
| Birth weight | Small | 46.8 | 51.3 | 33.0 | 4.5 | -18.3 | -13.8 |
| | Appropriate | 33.1 | 25.1 | 21.4 | -8 | -3.7 | -11.7 |
| | Large | 36.2 | 32.1 | 31.0 | -4.1 | -1.1 | -5.2 |

birth interval of more than 24 months than those born with less than 24 months interval in 2005 (AOR [95%CI] = 0.44[0.21, 0.92]) (Table 5).

## Multivariate decomposition analysis of selected predictors in the difference of neonatal mortality

The decomposition analysis shows that the increased birth intervals contributed to a 26% decline in the neonatal mortality rate between 2005 and 2016. This means that keeping the birth interval constant at the 2005 level would have raised the neonatal mortality rate by 26%

**Table 4.** Neonatal mortality rate (NMR) (five-year rate) and the difference in NMR between surveys, by selected maternal health-seeking behavioral factors, EDHS 2005, 2011 and 2016.

| **Variables** | | EDHS 2005 | EDHS 2011 | EDHS 2016 | 2011–2005 | 2016–2011 | 2005–2016 |
|---|---|---|---|---|---|---|---|
| | | NMR | NMR | NMR | NMR difference between surveys | | |
| Antenatal visits during pregnancy | no visit | 27.8 | 25.0 | 27.9 | -2.8 | 2.9 | 0.1 |
| | 1–4 visits | 35.4 | 27.6 | 18.5 | -7.8 | -9.1 | -16.9 |
| | ≥ 5 visits | 35.1 | 27.0 | 11.6 | -8.1 | -15.4 | -23.5 |
| Place of delivery | Home | 37.7 | 35.2 | 26.1 | -2.5 | -9.1 | -11.6 |
| | Health Inst | 57.8 | 51.1 | 34.0 | -6.7 | -17.1 | -23.8 |
| | Others | 49.6 | 125.7 | 43.0 | 76.1 | -82.7 | -6.6 |
| Mode of delivery | Vaginal | 38.9 | 36.9 | 27.3 | -2 | -9.6 | -11.6 |
| | C- section | 67.4 | 54.7 | 83.4 | -12.7 | 28.7 | 16 |
| Time postnatal check took place | within 24hrs | 145.6 | 100.8 | 27.8 | -44.8 | -73 | -117.8 |
| | after 24hrs | 1.9 | 8.0 | 1.1 | 6.1 | -6.9 | -0.8 |

**Table 5. Adjusted effects of selected socio-demographic, health seeking and fertility factors, among children born in the five years preceding the surveys, EDHS 2005, 2011 and 2016.**

| Variables | | EDHS 2005 | | | | EDHS 2011 | | | | EDHS 2016 | | | |
|---|---|---|---|---|---|---|---|---|---|---|---|---|---|
| | | AOR | 95% Cl | | P value | AOR | 95% Cl | | P value | AOR | 95% Cl | | P value |
| | | | LB | UB | | | LB | UB | | | LB | UB | |
| Regional Context | Agrarian | 1.00 | | | | 1.00 | | | | 1.00 | | | |
| | pastoralist | 0.59 | 0.27 | 1.30 | 0.18 | 0.58 | 0.26 | 1.30 | 0.22 | 1.38 | 0.61 | 3.10 | 0.48 |
| | City | 0.60 | 0.14 | 2.53 | 0.53 | 0.12 | 0.02 | 0.73 | 0.02 | 0.70 | 0.18 | 2.78 | 0.62 |
| Educational attainment | No Edu | 1.00 | | | | 1.00 | | | | | | | |
| | Primary | 0.49 | 0.17 | 1.44 | 0.19 | 1.53 | 0.44 | 3.03 | 0.39 | 1.70 | 0.44 | 6.63 | 0.44 |
| | Sec and higher | 1.00 | | | | 0.06 | 0.01 | 0.46 | 0.02 | 7.02 | 0.55 | 89.96 | 0.13 |
| Wealth status | poor | 1.00 | | | | 1.00 | | | | 1.00 | | | |
| | middle | 1.66 | 0.80 | 3.45 | 0.16 | 0.63 | 0.19 | 2.05 | 0.42 | 1.16 | 0.36 | 3.76 | 0.79 |
| | rich | 1.12 | 0.50 | 2.51 | 0.46 | 0.71 | 0.24 | 2.08 | 0.49 | 0.33 | 0.03 | 3.18 | 0.34 |
| Age at first birth | <18 | 1.00 | | | | 1.00 | | | | 1.00 | | | |
| | 18–34 | 0.56 | 0.26 | 1.23 | 0.10 | 0.94 | 0.42 | 2.44 | 0.85 | 0.93 | 0.26 | 1.65 | 0.88 |
| | ≥35 | | | | | | | | | | | | |
| Antenatal visits during pregnancy | no visit | 1.00 | | | | 1.00 | | | | 1.00 | | | |
| | 1–4 visits | 1.01 | 0.45 | 2.28 | 0.82 | 1.80 | 0.74 | 4.38 | 0.17 | 0.38 | 0.16 | 0.89 | 0.10 |
| | ≥5 visits | 0.10 | 0.01 | 0.81 | 0.03 | 0.51 | 0.08 | 3.18 | 0.08 | 0.10 | 0.02 | 0.60 | 0.01 |
| Place of delivery | Home | 1.00 | | | | 1.00 | | | | 1.00 | | | |
| | Health Inst | 1.83 | 0.37 | 9.02 | 0.41 | 0.74 | 0.13 | 4.25 | 0.74 | 1.42 | 0.59 | 6.61 | 0.55 |
| | others | 1.00 | | | | 1.00 | | | | 1.00 | | | |
| Preceding birth interval | <24mths | 1.00 | | | | 1.00 | | | | 1.00 | | | |
| | ≥24mths | 0.48 | 0.24 | 0.98 | 0.03 | 1.68 | 0.48 | 5.92 | 0.53 | 2.12 | 0.62 | 7.85 | 0.23 |
| Early initiation of BF | yes | 1.00 | | | | 1.00 | | | | 1.00 | | | |
| | no | 1.27 | 0.72 | 3.43 | 0.57 | 0.35 | 0.12 | 1.01 | 0.06 | 1.50 | 0.44 | 5.10 | 0.51 |
| Birth weight | Large | 1.00 | | | | 1.00 | | | | 1.00 | | | |
| | appropriate | 0.76 | 0.31 | 1.64 | 0.45 | 0.43 | 0.14 | 1.23 | 0.14 | 1.89 | 0.58 | 6.14 | 0.28 |
| | Small | 0.97 | 0.37 | 2.18 | 0.87 | 1.01 | 0.38 | 2.69 | 0.92 | 2.12 | 0.65 | 7.06 | 0.21 |

between 2005 and 2016. Similarly, a compositional change toward initiation of breastfeeding within one hour (early initiation breastfeeding), which is an improvement in the percentage of early initiation of breastfeeding, could decrease neonatal mortality by 10% between 2005 and 2016. Also, the model showed that the change in the effect of an individual intervention/predictor that individual children received did not contribute to the decrement in neonatal mortality rate between 2005 and 2016 (Table 6).

## Discussion

The study used nationally representative population-based survey data to analyze and identify the predictors of neonatal mortality and to estimate their contribution to the observed mortality reduction over 15 years in Ethiopia. Multivariate analysis showed that regional context, where newborns lived, and maternal ANC utilization were determinants for neonatal mortality in Ethiopia. Decomposition analysis also indicated that early initiation of breastfeeding and birth interval contributed to reducing neonatal mortality in the country during the survey period.

The study showed that neonatal mortality in Ethiopia remains higher than the average for the sub-Saharan region, which is 27 per 1,000 live births, according to the World Health

**Table 6. Multivariate decomposition of selected socio demographic, health seeking, fertility factors related differences in the NMR, EDHS 2005, 2011, and 2016.**

| Neonatal death | Due to Difference in Characteristics (E) | | | | | Due to Difference in Coefficients (C) | | | | |
|---|---|---|---|---|---|---|---|---|---|---|
| | Coef. | 95% Conf. Interval | | P value | Pct (%) | Coef. | 95% Conf. Interval | | Pct (%) | P value |
| | | LB | UB | | | | LB | UB | | |
| Regional Context | | | | | | | | | | |
| Agrarian | 0.000034692 | -0.0017879 | 0.0018572 | 0.97 | 2.00 | 0.001398 | -0.014243 | 0.01704 | 80.63 | 0.86 |
| pastoralist | 0.00004363 | -0.0016642 | 0.0017516 | 0.96 | 2.51 | 0.001233 | -0.011654 | 0.014122 | 71.14 | 0.85 |
| City | 0 | 0 | 0 | | 0 | 0 | 0 | 0 | | |
| Educational attainment | | | | | | | | | | |
| No Edu | 0.00022787 | -0.0015236 | 0.0019794 | 0.79 | 13.13 | 0.001239 | -0.023642 | 0.026121 | 71.45 | 0.92 |
| Primary | -0.00012728 | -0.0016354 | 0.0013809 | 0.86 | -7.34 | -0.00099 | -0.011094 | 0.0091072 | -57.28 | 0.85 |
| Sec and higher | 0 | 0 | 0 | | 0 | 0 | 0 | 0 | | |
| Wealth status | | | | | | | | | | |
| poor | 0.00042714 | -0.000363 | 0.0012173 | 0.29 | 24.63 | -0.005002 | -0.038211 | 0.028206 | -288.4 | 0.76 |
| middle | 0.00016142 | -0.0001753 | 0.0004981 | 0.35 | 9.31 | -0.00013 | 0.0032853 | 0.003026 | -7.46 | 0.94 |
| Rich | 0 | 0 | 0 | | 0 | 0 | 0 | 0 | | |
| Age at first birth | | | | | | | | | | |
| <18 | 0.00031339 | 2.2317E-05 | 0.0006491 | 0.07 | 18.07 | 0.00074 | 0.0060741 | 0.007555 | 42.72 | 0.83 |
| $\geq$ 18 | 0 | 0 | 0 | | 0 | 0 | 0 | 0 0 | | |
| Antenatal visits during pregnancy | | | | | | | | | | |
| no visit | 0.001751 | -0.0014911 | 0.0049931 | 0.29 | 100.9 | 0.00727 | -0.056888 | 0.042335 | -419.54 | 0.77 |
| 1–4 visits | 0.00039175 | -0.0019741 | 0.0027577 | 0.75 | 22.59 | 0.00389 | -0.025991 | 0.033787 | 224.76 | 0.79 |
| $\geq$ 5 visits | 0 | 0 | 0 | | 0 | 0 | 0 | 0 0 | | |
| Place of delivery | | | | | | | | | | |
| home | 0.00031339 | -0.0031629 | 0.0030112 | 0.96 | -4.37 | 0.011685 | -0.063629 | 0.086999 | 673.78 | 0.76 |
| Health Inst | 0 | 0 | 0 | | 0 | 0 | 0 | 0 0 | | |
| Preceding birth interval | | | | | | | | | | |
| <24 months | -0.00045579 | -0.0007326 | -0.0001790 | 0.01 | -26.28 | 0.0034131 | -0.018003 | 0.024829 | 196.8 | 0.75 |
| $\geq$24 months | 0 | 0 | 0 | | 0 | 0 | 0 | 0 0 | | |
| Early initiation of BF | | | | | | | | | | |
| Yes | 0.00017701 | 8.2654E-05 | 0.0002713 | <0.01 | 10.21 | 0.00077 | -0.013706 | 0.015253 | 44.58 | 0.92 |
| No | 0 | 0 | 0 | | 0 | 0 | 0 | 0 0 | | |
| Birth weight | | | | | | | | | | |
| Large | 4.08E-08 | 1.2657E-07 | 2.08E-07 | 0.63 | 0.01 | -0.00266 | -0.020548 | 0.015224 | -153.5 | 0.77 |
| Average | 0.00002141 | 2.2607E-06 | 0.0000450 | 0.07 | 1.23 | 0.000732 | 0.0085547 | 0.01002 | 42.23 | 0.88 |
| Small | 0 | 0 | 0 | | | 0 | 0 | 0 0 | | |
| Over all | .0028905 | -.00079551 | .0065765 | 0.12 | 166.6 | -.001156 | -.0073623 | .0050499 | -66.67 | 0.71 |

Organization [2]. This is despite the fact that the country has implemented community based and essential newborn care interventions under the national newborn and child survival strategy to overcome neonatal mortality for two decades. Community based newborn care aimed at actively engaging communities in promoting and preventing newborn healthcare through health extension workers (HEWs) and health development army's (HDAs) at the community level. Essential newborn care services are services addressing the specific needs of newborns from birth up to 28 days and provided at health facilities level [28]. We observed that between 2005 and 2016, the country's mortality rate was reduced by 11 per 1,000 live births, from 39 per 1000 live births in 2005 to 28 per 1000 in 2016, but still there is a need to improve neonatal mortality to meet the SDG3 target of <12 per 1,000 live birth. There were also differences in

neonatal mortality between regions of the country; for example, neonatal mortality has increased in the pastoralist region over the past 15 years. As a result, children born in the pastoralist region have the highest neonatal mortality compared to those born in the agrarian region and cities. This finding is supported by previous studies conducted in the pastoralist regions of Ethiopia [29, 30]. The potential reason for the finding is that the absence of health service delivery systems that are tailored to pastoralist lifestyles and setting may amplify the number of unmet demands and deter women in pastoralist communities from seeking health-care during pregnancy and childbirth, resulting in low utilization of maternal and child health-care, and contributing to high neonatal mortality in the regions [31]. Furthermore, low vaccine coverage and high incidence of infectious diseases in pastoralist regions leads to neonatal mortality to get worse in the regions [30].

Newborns whose mothers had antenatal care visits during pregnancy had lower odds of dying in the first month of life. This finding is supported by other studies conducted in Kenya and Uganda [32] and a meta-analysis among sub-Saharan Africa studies [8]. The possible mechanism for this finding is that mothers who had ANC visits received health information from health care providers about healthy behaviors and possible medical complications during and after pregnancy, including newborn care. Also, the mothers are screened for health problems and receive curative and preventive health care services, such as tetanus immunization and iron and folic acid supplementation, to improve newborn health [33].

Another significant predictor of neonatal mortality was the previous birth interval. Neonatal mortality was higher among children with a preceding birth interval of less than 24 months. Similar findings have been reported in Nigeria [34] and Bangladesh [15]. The possible explanation for this finding could be that mothers with birth intervals of less than 24 months might not get enough time to replenish their nutritional status that had been depleted from the previous pregnancy [14]. Poor maternal nutritional status during pregnancy may affect fetal growth, leading to poor newborn health outcomes. Also, mothers with short birth intervals might not go for ANC visits due to their young children, resulting in mothers not getting the recommended maternal healthcare services during pregnancy, contributing to an unfavorable health status for the newborns.

The decomposition analysis indicated that the preceding birth interval had a significant effect on the reduction of neonatal mortality. The decrease in number of women with a birth interval of less than 24 months, resulted in a decline in neonatal mortality in the survey period. Early initiation of breastfeeding was another endowment factor that significantly reduced neonatal mortality. The increase in the prevalence of early initiation of breastfeeding is related to a reduction in neonatal mortality between 2005 and 2016. The possible reason might be that early initiation of breastfeeding could decrease the risk of newborns ingesting infectious agents, reducing the infection that causes neonatal mortality. Besides, the first breastmilk, colostrum, is full of immunoglobulin and lymphocytes, stimulating the newborns' immune systems to prevent infection [12]. The finding is consistent with prior studies on newborn survival and early initiation of breastfeeding in Ghana [35] and India [13].

Neonatal mortality can be preventable, but it remains a major public health issue in Ethiopia, particularly in pastoralist communities, due to low access and utilization maternal and child healthcare services in pastoralist regions. This emphasized the importance of improving access and barriers to maternal healthcare services to reduce neonatal mortality in the regions. Therefore, efforts are needed to prevent neonatal mortality that targets pastoralist communities. Besides, reducing preventable neonatal mortality in pastoralist communities can help the country achieve national and global targets. It is also an essential target of the Sustainable Development Goals.

Among the potential limitations of this study, there is a possibility of recall bias from DHS surveys on events that happened in the past and based on other retrospective data. The study used secondary data, and there might be an unexplained association among variables due to confounders. Finally, it is a cross-sectional study, making it difficult to establish a causal relationship between predictors and neonatal mortality.

## Conclusion

The study revealed that neonatal mortality is still a major public health concern in Ethiopia. There is a need to strengthen a tailor-made maternal and child healthcare services that consider the pastoralist regions context, to enhance the access and utilization of maternal and child healthcare services in the pastoralist communities. Moreover, early breastfeeding initiation and a birth interval of more than 24 months are crucial preventive measures to reduce neonatal mortality. Thus, healthcare providers should educate mothers about these measures during their ANC visits to ensure newborn health.

## Acknowledgments

We appreciate the contributions of DHS program staff technical assistance.

## Author Contributions

**Conceptualization:** Yirgalem Shibiru Baruda, Mark Spigt.

**Formal analysis:** Yirgalem Shibiru Baruda.

**Project administration:** Yirgalem Shibiru Baruda.

**Supervision:** Mark Spigt, Andrea Gabrio, Lelisa Fikadu Assebe.

**Writing – original draft:** Yirgalem Shibiru Baruda.

**Writing – review & editing:** Mark Spigt, Andrea Gabrio, Lelisa Fikadu Assebe.

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
