## [Decision Letter · Decision Letter 0]

12 Dec 2023

PGPH-D-23-02158

Predictors of Neonatal Mortality in Ethiopia: Cross Sectional Study Using 2005, 2010 and 2016 Ethiopian Demographic Health Survey Demographic Health Survey Datasets

Dear Dr. Baruda,

Thank you for submitting your manuscript to PLOS Global Public Health. After careful consideration, we feel that it has merit but does not fully meet PLOS Global Public Health’s publication criteria as it currently stands. Therefore, we invite you to submit a revised version of the manuscript that addresses the points raised during the review process.

We look forward to receiving your revised manuscript.

Kind regards,

Vinay Nair Kampalath, MD, DTMH

Guest Editor

Journal Requirements:

3. Please include a title page at the beginning of your manuscript file that lists full author names and institute addresses.  This should not be uploaded as a separate file.  

Additional Editor Comments (if provided):

Please do respond to the comments and revisions suggested by the two reviewers. We invite you to resubmit after the comments/revision suggestions have been addressed.

Reviewers' comments:

Reviewer's Responses to Questions

**Comments to the Author**

1. Does this manuscript meet PLOS Global Public Health’s publication criteria? Is the manuscript technically sound, and do the data support the conclusions? The manuscript must describe methodologically and ethically rigorous research with conclusions that are appropriately drawn based on the data presented.

Reviewer #1: Yes

Reviewer #2: Partly

2. Has the statistical analysis been performed appropriately and rigorously?

Reviewer #1: I don't know

Reviewer #2: I don't know

3. Have the authors made all data underlying the findings in their manuscript fully available (please refer to the Data Availability Statement at the start of the manuscript PDF file)?

Reviewer #1: Yes

Reviewer #2: Yes

4. Is the manuscript presented in an intelligible fashion and written in standard English?

Reviewer #1: Yes

Reviewer #2: Yes

5. Review Comments to the Author

Reviewer #1: Please see attached file. Main concerns include:

1. Cited reference does not confirm Ethiopia has the highest NMR in the world

2. Some clarification of language

3. More explanation and/or clarification for discrepant data is requested

4. Numbering of tables is incorrect

Reviewer #2: The article is well prepared, it needs minor edits on most of the points. I storngly suggest that the results presented are too much and crowded to viewwers, if you can work on that and make it easy to look and understand, also if you can use alternative methods other than table, that would make it better.

The other strong comment i have is on the conclusion, It needs to be done again, for me it lost the strenght when i reach to the conclusion. Also if you can include recommendation that can be applicabel it would be a good contribution, since you used a national data.

6. PLOS authors have the option to publish the peer review history of their article (what does this mean?). If published, this will include your full peer review and any attached files.

**Do you want your identity to be public for this peer review?** For information about this choice, including consent withdrawal, please see our Privacy Policy.

Reviewer #1: No

Reviewer #2: No

---

## [Decision Letter · Decision Letter 1]

14 Feb 2024

Predictors of Neonatal Mortality in Ethiopia: Cross Sectional Study Using 2005, 2010 and 2016 Ethiopian Demographic Health Survey Demographic Health Survey Datasets

PGPH-D-23-02158R1

Dear Mr Baruda,

We are pleased to inform you that your manuscript 'Predictors of Neonatal Mortality in Ethiopia: Cross Sectional Study Using 2005, 2010 and 2016 Ethiopian Demographic Health Survey Demographic Health Survey Datasets' has been provisionally accepted for publication in PLOS Global Public Health.

Best regards,

Vinay Nair Kampalath, MD, DTMH

Guest Editor

Thank you for addressing the reviewers' comments. We accept this manuscript for publication.

Reviewer Comments (if any, and for reference):

Reviewer's Responses to Questions

**Comments to the Author**

1. If the authors have adequately addressed your comments raised in a previous round of review and you feel that this manuscript is now acceptable for publication, you may indicate that here to bypass the “Comments to the Author” section, enter your conflict of interest statement in the “Confidential to Editor” section, and submit your "Accept" recommendation.

Reviewer #1: All comments have been addressed

Reviewer #2: All comments have been addressed

2. Does this manuscript meet PLOS Global Public Health’s publication criteria? Is the manuscript technically sound, and do the data support the conclusions? The manuscript must describe methodologically and ethically rigorous research with conclusions that are appropriately drawn based on the data presented.

Reviewer #1: Yes

Reviewer #2: Partly

3. Has the statistical analysis been performed appropriately and rigorously?

Reviewer #1: I don't know

Reviewer #2: I don't know

4. Have the authors made all data underlying the findings in their manuscript fully available (please refer to the Data Availability Statement at the start of the manuscript PDF file)?

Reviewer #1: Yes

Reviewer #2: Yes

5. Is the manuscript presented in an intelligible fashion and written in standard English?

Reviewer #1: Yes

Reviewer #2: Yes

6. Review Comments to the Author

Reviewer #1: (No Response)

Reviewer #2: The Authors have addressed almost all of my previous comments. For which they haven't they have given a reasonable explanation.

7. PLOS authors have the option to publish the peer review history of their article (what does this mean?). If published, this will include your full peer review and any attached files.

**Do you want your identity to be public for this peer review?** For information about this choice, including consent withdrawal, please see our Privacy Policy.

Reviewer #1: No

Reviewer #2: No
